# Evaluation of Remote Sensing Ecological Index Based on Soil and Water Conservation on the Effectiveness of Management of Abandoned Mine Landscaping Transformation

**DOI:** 10.3390/ijerph19159750

**Published:** 2022-08-08

**Authors:** Zeke Lian, Huichao Hao, Jing Zhao, Kaizhong Cao, Hesong Wang, Zhechen He

**Affiliations:** 1School of Landscape Architecture, Beijing Forest University, Beijing 100083, China; 2School of Theater, Film and Television, Communication University of China, Beijing 100024, China; 3School of Ecology and Nature Conservation, Beijing Forest University, Beijing 100083, China; 4College of Forestry, Beijing Forest University, Beijing 100083, China

**Keywords:** land ecology, mine management, remote sensing ecological index, ecological restoration assessment, soil and water conservation

## Abstract

Abandoned mines are typical areas of soil erosion. Landscape transformation of abandoned mines is an important means to balance the dual objectives of regional ecological restoration and industrial heritage protection, but the secondary development and construction process of mining relics require long-term monitoring with objective scientific indicators and effective assessment of their management effectiveness. This paper takes Tongluo Mountain Mining Park in Chongqing as an example and uses a remote sensing ecological index (RSEI) based on Landsat-8 image data to assess the spatial and temporal differences in the dynamic changes in the ecological and environmental quality of tertiary relic reserves with different degrees of development and protection in the park. Results showed that: ① The effect of vegetation cover, which can significantly improve soil and water conservation capacity. ② The RSEI is applicable to the evaluation of the effectiveness of ecological management of mines with a large amount of bare soil areas. ③ The mean value of the RSEI in the region as a whole increased by 0.090, and the mean values of the RSEI in the primary, secondary and tertiary relic reserves increased by 0.121, 0.112 and 0.006, respectively. ④ The increase in the RSEI in the study area is mainly related to the significant decrease in the dryness index (NDBSI) and the increase in the humidity index (WET). The remote sensing ecological index can objectively reflect the difference in the spatial and temporal dynamics of the ecological environment in tertiary relic protection, and this study provides a theoretical reference for the ecological assessment of secondary development-based management under difficult site conditions.

## 1. Introduction

Ecological restoration and environmental protection management of mines are currently the focus of global attention [1] and is a global concern. As a special landscape remnant of the human transformation of nature, mining relics can be transformed into parks by means of landscape transformation [2]. This study examines the transformation of abandoned mines into parks and open spaces, which can realize the protection of industrial relics [3] and natural ecological restoration [4] and the economic development of the surrounding area [5,6] and other Multiple goals [7,8]. However, the process of “turning mines into parks” requires secondary development and construction of the site [2,9,10], and long-term monitoring of mine ecological restoration is necessary to avoid negative impacts on regional ecology [11].

Detailed evaluation of sample sites, is constrained by spatial and temporal conditions [12,13,14,15,16]. It is difficult to form a comprehensive long-term evaluation. With its multitemporal, high-spatial coverage and easy and rapid access, remote sensing technology has become an important tool for the long-term monitoring of regional ecological changes [17,18,19,20]. However, most related work is based on basic observations using single indicators [21]. However, most work is based on basic observations using single indicators, especially vegetation indicators [22]. In 2013, Xu Hanqiu proposed the remote sensing ecological index (RSEI), which is a measure of the ecological change in the region [23]. In 2013, Xu Hanqiu proposed the remote sensing ecological index (RSEI) as a monitoring and evaluation tool for regional ecological conditions and a rapid detection and evaluation of regional ecological long time series [24]. The four items of the RSEI are among the four ecological factors of the RSEI. The heat index and dryness index obtained from the inversion of surface temperature and bare soil index have a high contribution rate in the evaluation of the ecological quality of mines and are closely related to human life [25,26]. The RSEI has more comprehensive and social criteria applicability, which makes up for the deficiency of the traditional evaluation method of using a normalized vegetation index to evaluate ecological quality. Existing studies based on the RSEI have conducted a large number of studies on cities and urban clusters [27,28,29], nature reserves [30,31,32], soil erosion areas [33,34], and mines [35].

Tongluo Mountain Mine Park in Chongqing adopts the idea of “turning mines into gardens” and limits the degree of protection and development of the 41 quarry relics in the park to three levels, fully reflecting the impact of different degrees of secondary development and construction on ecological restoration. Here we asked: (1) Can the ecological quality of the study area be improved by the ecological treatment of “mine to park”, which requires secondary development and construction? (2) What are the spatial and temporal dynamics of the ecological environment of each quality level in the three-level protected areas with different degrees of development and protection? (3) Can the remote sensing ecological index (RSEI) reflect the differences in the effectiveness of ecological management in the study area under different levels of development and management? This study is expected to enrich the evaluation system of ecological restoration effectiveness in mining areas and has long-term practical significance for ecological construction in the study area.

## 2. Materials and Methods

### 2.1. Materials

Located in Shifun town and Yupeng Mountain town, Yubei District, Chongqing Tongluo Mountain Mining Park was once the largest limestone quarry group in Yubei District, and since the 1980s, large-scale gravel-mining activities have been carried out in Tongluo Mountain. By 2010, long-term mining had left 41 pits of various shapes and forms, forming an impact area of 14.87 square kilometers. The quarrying area stretched 10 km from north to south, with bare rock exposed in the pits and significant degradation of vegetation around the pits, exposing the local ecological environment to great safety hazards. From 2010 to 2012, the city of Chongqing completely shut down the Tongluo Mountain quarry, and 12 water pits gradually formed in the pit complex due to groundwater activities, with minerals dissolving in the water forming a unique turquoise blue pit lake with unique scenery, attracting many surrounding residents. The unique scenery attracts many residents to visit this place.

Based on the special landscape of Tongluo Mountain, the local government has carried out ecological management of the quarry area in phases since 2014 and has built Tongluo Mountain Mine Park, which has a total planning area of 2472.25 hm^2^, of which 691.75 hm^2^ is the core area, 171.47 hm^2^ is the scope of the heavy control area, and 2300.78 hm^2^ is the scope of the no-build area. The park has 41 limestone pit relics, and according to the typicality, rarity and ornamental nature of the mining relics, as well as their scientific, historical and cultural values and development and utilization functions, the mining relics in Tongluo Mountain are graded and protected as rare (Grade 1), important (Grade 2) and general (Grade 3). Each level is separately designated as a protected area in accordance with the relevant requirements, and the protection requirements are determined.

The delineation of the protected areas is mainly based on the level of protection of the relics, which is shown in the Figure 1. Within a radius of 100 m of the Grade I mining remains, the zone is designated as a Grade I protection zone. Construction is prohibited within the zone, and the original appearance of the mining ruins is maintained as much as possible. Within a 50-m radius of the secondary mining relics, the zone is designated as a secondary protection zone, where construction is prohibited, and scientific activities are carried out in the form of cultural experiences in trace camps. Within a 20-m radius of the tertiary mining ruins is designated as a tertiary protection zone, with moderate development and construction within the tertiary protection zone, with cultural buildings and necessary tourism service facilities arranged around the theme of mine science education.

### 2.2. Methods

#### 2.2.1. Data Sources

Landsat has a large amount of historical data compared with other satellite data, which is conducive for the study of long-time series of the ground surface. Therefore, this paper uses Landsat remote sensing images as the data source to analyze and evaluate the ecological environment of Tongluo Mountain Mining Park in Chongqing. The data were obtained from the United States Geological Survey (USGS) and Geospatial Data Cloud. To ensure that the surface ecology was not different due to seasonal differences, remote sensing data were selected from July to August; when it was summer, the vegetation growth condition was good, which is conducive to the evaluation study of ecological quality of the mine area. In this paper, Landsat 8 TM images from July 2014, August 2019 and August 2021 were used. The selected data all meet the characteristics of high data quality and low cloud content, and the image quality meets the research needs. The remote sensing images were preprocessed in ENVI5. 3 software for radiometric calibration and atmospheric correction, and finally, the remote sensing images were cropped based on the boundary of Tongluo Mountain Mining Park.

#### 2.2.2. Research Methodology

The remote sensing ecological index (RSEI) is a more mature model in the study of ecological status evaluation. The index is entirely based on the information of remote sensing image data, and the four index factors of vegetation, heat, humidity, and dryness are obtained by inversion of remote sensing image data. The vegetation index is represented by the normalized differential vegetation index (NDVI), which is commonly used to monitor vegetation growth and vegetation cover; the moisture index is represented by the moisture component (WET), which is related to soil moisture in the tassel cap transformation; and the moisture component (WET), which is related to soil moisture in the tassel cap transformation, is used to represent the moisture index, which is a response to the water content of the surface. Land surface temperature (LST), an important parameter reflecting the energy flow and material exchange in the soil–vegetation–atmosphere system, represents the heat index; the index-based built-up index (IBI) and soil index (SI), which jointly influence surface dryness, are taken as the average values to represent the dryness index. The formulae for calculating the index factors are as follows:(1)The normalized difference vegetation index (NDVI) was used to represent the greenness component (also called greenness index), which was modeled as:
(1)$$\mathrm{NDVI}=\left(\rho_{nir}-\rho_{red}\right)/\left(\rho_{nir}+\rho_{red}\right)$$(2)Humidity component WET based on tassel cap variation is sensitive to humidity, $\rho_{i}$ indicates the spectral reflectance of the corresponding waveband:(2)$$\mathrm{WET}=0.1511\rho_{blue}+0.1973\rho_{green}+0.3283\rho_{red}+0.3407\rho_{nir}\;-0.7117\rho_{mir1}-0.4559\rho_{mir2}$$(3)The NDBSI is expressed as the average of the exponential building index (IBI) and the bare earth index (SI):
(3)NDBSI=(IBI+SI)/2(4)The heat index is expressed in terms of surface temperature LST, and the other indices are calibration parameters:
(4)$$\mathrm{LST}=T/\left[1+\left(\frac{\lambda T}{\rho }\right)ln\varepsilon \right]$$

The RSEI coupling the four indicators were constructed. Because of the differences in the values of the indicators, standardization and dimensionless processing were performed, and the calculation formula is:(5)$$\mathrm{NX}=\frac{X-X_{min}}{X_{max}-X_{min}}$$

In the formula, NX is the result of the normalization of the index, *X* is the mean of the image elements of this index, *X_max_* and *X_min_* are the maximum and minimum values of the indicator, respectively.

In data mining-related applications, principal component analysis is one of the most common methods, and its key advantage is that the weights of each indicator are non-artificially determined, which can reduce the errors caused by human interference. Principal component analysis is used for index integration of remote sensing ecological indices, and the initial value is obtained by using Envi software, and the initial value $RSEI_{0}$ is obtained by subtracting PC1 from 1, and then it is standardized, and the resulting RSEI is the remote sensing ecological index, calculated by the formula:(6)$$\mathrm{RSEI}=\frac{RSEI_{0}-RSEI_{0\_min}}{RSEI_{0\_max}-RSEI_{0\_min}}$$

In the formula, *RSEI*_0_ is the initial remotely sensed ecological index, *RSEI*_0_*max*_ and *RSEI*_0_*min*_ are the maximum and minimum values of the initial remote sensing ecological index, respectively, and the RSEI is the final remote sensing ecological index with a value range of [0, 1]; the larger the value is, the better the ecological environmental quality.

## 3. Results

### 3.1. Results of Principal Component Analysis

As seen from Table 1, the contribution rates of the eigenvalues of the first principal component PC1 are all up to more than 80%, i.e., the first principal component concentrates the characteristic information of the four subindicators to the maximum extent. In PC1 of the results of the principal component analysis in each year, WET and NDVI are positive, and NDBSI and LST are negative, which is consistent with the general perception of the feedback relationship between the four indicators and the ecological environmental quality, namely, moisture and vegetation cover have positive effects on ecological environmental quality, and the degree of surface exposure, anthropogenic floor area and surface temperature have negative effects on ecological environmental quality. Among them, the highest value of green load indicates that the vegetation factor has the greatest influence on ecological environmental evaluation, which means that vegetation has the greatest effect on environmental quality improvement. Meanwhile, the sum of the eigenvalues of the vegetation index (NDVI) and humidity index (WET) are smaller than the sum of the absolute values of the eigenvalues of the dryness index (NDBSI) and heat index (LST), indicating that the improvement effect of vegetation and humidity on ecology is weaker than the damage effect of dryness and heat on ecology. Among them, the vegetation index (NDVI) and the humidity index (WET) respond to the local green cover area, which has a significant effect on local cooling and humidification, while the plants have good soil and water conservation ability for areas with large slopes such as mine pits. At the same time, the humidity index (WET) can also respond well to the enhancement of soil and water conditions.

As seen from Table 2, the RSEI values in the study area improved considerably from 2014 to 2021, with an overall increase of 0.089948, and the overall ecological quality entered a good grade. The percentage of excellent ecological grade increased by 16.14%, and the percentage of poor and poorer grade areas decreased by 12.22%. The ecological quality of the overall study area showed an upward trend. In recent years, with restoration work, the ecological quality of the mine area has shown significant improvement, mainly because the mine area has been treated in a three-level way, and different treatment measures have been adopted for different areas, resulting in different index changes. Through an analysis of the changes in the four ecological factors, the remote sensing ecological index constructed by the factors shows good applicability in the ecological quality monitoring of the limestone mine area and can reflect the ecological characteristics of different graded treatment areas.

### 3.2. Analysis of the Ecological Quality and Soil and Water Conservation Capacity of the Study Area

A grading map of the remote sensing ecological index was made (Figure 2a,b). The area of each grade and the proportions were determined (Table 2).

Between 2014 and 2021, the overall ecological quality within the scope of Tongluo Mountain Mine Park improved generally, and the remote sensing ecological index increased by 0. 028. Its ecological management effectiveness was mainly manifested by an increase of 1.10 km^2^ in the area of excellent ecological grade within the region, which increased from 4.05% to 20.19%, and the area of excellent grade ecological quality within the study area increased significantly, reflecting the ecological management effects. The area of excellent ecological quality in the study area increased significantly, reflecting the effectiveness of ecological management and the lag of results with typical performance of vegetation restoration, a large number of vegetation restoration areas are mainly located in the original bare soil areas around the mine pits, which greatly enhance the soil and water conservation capacity of the vegetation covered areas and strongly prevent the erosion of the original landforms by heavy rainfall; meanwhile, the area of poor ecological grade decreased by 9.05%, among which the area of poorer grade decreased by 3.17%. During the treatment period, a large number of bare soil areas reclaimed from mines was restored, which had a significant effect on the ecological quality improvement and RSEI. The large number of re-greened areas has raised the humidity level of the area and most of the pits have formed a cultured water source in the center, enhancing the surrounding hydrological conditions. After the restoration, the low-quality areas at this stage were mainly distributed on both sides inside and outside the mining relic reserve: within the reserve, the construction of the landscape facilities of the mine ecological park and the space of buildings and squares in the park formed part of the low-quality ecological areas, which can be seen in the Figure 3.

### 3.3. Analysis of Ecological Dynamic Changes in the Study Area

ENVI software was used to detect the change information of the RSEI from 2014 to 2021 and generate the ecological change map (Figure 2c). From Table 3, we can see that the percentage of the area with no ecological change between 2014 and 2021 was 0%, and the overall ecological quality of the study area had a large range of change; the percentage of the area with ecological improvement was 77.88%, and the percentage of the area with only one ecological grade improvement was 69.89%. The ecological quality was improved mainly in the areas around the mine pits within the study area due to the mine pit protection project, while ecological quality was degraded in 22.12% of the area, with 19.77% of the ecological quality reduced by one grade and 2.12% of the ecological quality reduced by two grades or more. The ecological quality reduction area was mainly concentrated around Pits 1–8. This indicates that large-scale construction has had a significant negative impact on the ecological restoration effects around the reserve (Pits 1–8), and the surrounding villages have been expanded and rebuilt to a certain extent, resulting in a significant decrease in the index around the pits. Although there was a certain decline in the index, the park construction of the mine pits has been completed, forming a good landscape ornamental effect, and to a certain extent, the conservation and utilization requirements of the mine pits in the plan have been completed.

### 3.4. Analysis of Changes in Ecological Quality and Soil and Water Conservation Capacity of Tertiary Conservation Areas

To further analyze the differences between the subtreatment measures, remotely sensed ecological indices (Table 4) were calculated separately within the three levels of treatment areas to reflect the degree of change in ecological quality within the areas and the role of each area in influencing the ecological quality of the whole park. From the change in the ecological quality index from 2014 to 2021, the value of the RSEI for the whole park improved by 0.089948 for all three graded areas. The primary area was a key restoration area with a low RSEI of 0.455299 in 2014, which was at a medium level, after which the area was treated more vigorously, and the overall area improved by 0.121361 to 0.576660 at the beginning and end of the treatment phase. The secondary area had a better ecological base, with a RSEI of 0.498493 in 2014, which was slightly better than the primary area but also at a medium level. The ecological quality of this area also had an increasing trend, with the overall area improving by 0.111994 at the beginning and end of the treatment phase, and the final RSEI reached 0.61487, rising to a good level. The tertiary area was at the stage of the lowest level of protection and could be built on a certain scale, taking up the main tour function, but the RSEI of the tertiary area reached 0.604446 in 2014, which is a good level, and the overall area improved by 0.006093 at the beginning and end of the treatment stage, and the RSEI reached 0.610539 after the treatment, which remained at a good level.

To further compare and analyze the characteristics of ecological quality changes among the three levels of areas between 2014 and 2021, the share of areas with different ecological grades within the three levels in 2014 and 2021 were counted (Figure 4), and the graphs show that the share of areas with an ecological quality index of 0.0–0.2 in the three graded areas decreased significantly totally 16.04% between 2014 and 2021, with the first-level areas. The reason for this is that the primary protected area, as the most severely mined area, has a large number of bare soil mining areas with the greatest risk of soil erosion, therefore it had the greatest protection and the best re-greening effect in the first stage. The re-greening effect is obvious in the location of the slope, the overall re-greening area is large, and the formation of the mine cultured water source to promote each other, playing a good cooling and humidification, reduce rainwater scouring effect of soil and water conservation; It is especially obvious in the RSEI. In contrast, the lowest ecological indices in secondary and tertiary areas also had a decreasing trend, but at the same time, the ecological quality of these two areas was better, the area of the ecological quality index 0.8–1.0 improved significantly under the protection measures, indicating that the secondary and tertiary areas showed a “better quality” ecological restoration effect after the implementation of the protection measures. Therefore, the three types of graded areas have adopted different restoration strategies to achieve a comparable post-restoration level in the face of different restoration status.

The ecological quality of all three regions improved during 2014–2021, but the three regions adopted different treatment measures; therefore, the reasons and characteristics of the ecological quality improvement in the three regions can be analyzed in detail based on the changes in the normalized index values of ecological factors in the four regions from 2014–2021 (Table 5 and Figure 5).

Among the graded regions in 2014, the ecological damage area of the tertiary region was smaller, the degree of damage was lower, and the ecological damage of the primary region was the greatest; therefore, in 2014, the dryness index of the primary region was highest, the secondary region was second, and the tertiary region was in the best condition. Figure 2a shows that there was some concentrated ecological damage within the primary region, and the large area of bare soil in the region was mainly due to the mining caused by mining. With the graded protection measures starting in 2014, the vegetation and moisture index within the primary area appeared to increase, and by 2021, the vegetation of the primary area increased by 0.028941, and the dryness index decreased by −0.126772. This stage of improvement was related to local high-intensity protection initiatives, meanwhile the large amount of vegetation cover effectively mitigates the risk of soil erosion in this area. At the same time, the formation of a large number of natural puddles in the primary area had a certain effect on the overall ecological quality improvement, and the humidity index increased by 0.222823 between 2014 and 2021. The four ecological factors in the index indicate that the primary area had importance for ecological restoration and the development and construction of conservation initiatives were not carried out, and the integrated RSEI also increased by 0.121361. The ecological base of the secondary area was better, and the vegetation index in 2014 reached 0.582730 higher than the vegetation value of the primary area after treatment in 2021, which was 0.098211, indicating that the vegetation status of the secondary area was significantly better than that of the primary area. At the same time, the dryness index of the secondary area reached 0.500751, and the destruction of bare soil was comparable to that of the primary area. The government adopted the approach of attaching importance to ecological restoration and trace landscape development for the secondary area and adopted high-intensity ecological restoration measures to increase the vegetation coverage area for the bare area and the damaged area in the protected area, while at the same time carrying out trace development for the scenic quality resources of the mine area to meet the visitation demand of the surrounding residents, but the development intensity was weak and did not cause an excessive impact on the ecological quality. By 2021, with the progress of conservation work, the vegetation index of the secondary area increased to 0.635925, which was 0.053195 higher than that in 2014, and the dryness index decreased by 0.118600, which indicates that the bare soil recovered better, and the humidity index increased by 0.150914, which was low compared with that of the primary area, because the natural puddles formed in the mine area in the secondary area were worse than those in the primary area. The Level I area was poor. As the best ecological substrate, the vegetation index of the tertiary area was as high as 0.604446 in 2014, which exceeded the pre-restoration level of the primary and secondary areas. Therefore, in addition to ecological restoration, the policy of protecting this area also undertook the construction of tour facilities and provided cultural tour services; therefore, there was a certain degree of engineering construction in the tertiary area, which caused a certain degree of negative impact on the ecosystem; and in terms of ecological factors, the increase in vegetation between 2014 and 2021 was only 0.005648 less than the other two areas, which was 19. 51% and 10. 32% of the other two areas, respectively. The decrease in the dryness index was only 0.037637, which was 29.69% and 31.73% of the other two regions, respectively. However, after the restoration of the tertiary area, not only the construction of several landscape tour facilities was carried out but also the improvement of ecological quality was completed.

## 4. Discussion

Rare earth minerals are a non-renewable resource, and are being more extensively used in machinery manufacturing, petrochemical industry, agriculture, forestry and animal husbandry, aerospace and military technology. Due to the increasing demand for rare earth resources, the mining scale of rare earth mines has been expanded, and some unscrupulous businessmen are mining rare earth mines beyond the mine area according to the scope for profiteering, and the improper mining methods will certainly lead to plant destruction, ecological quality decline, soil erosion and many other environmental problems in rare earth mines. Facing such problems, the landscape transformation of abandoned mines is an important measure to protect and reuse industrial heritage. In the face of different mine status quo, different restoration priorities and strategies are specified according to local conditions, and the ecological factor screening out strategy by remote sensing ecological index suppresses the risk of soil erosion and improves the ecological quality of the mine.

The Tongluo Mountain Mining Park mentioned in this paper divides the mining remains in the park into three levels (Figure 1) for graded control and protection to enhance ecological quality and reduce the risk of soil erosion. The primary protected area was only protected without development, and the area contains eight naturally occurring water mine pits. The significant improvement in its internal ecological quality over seven years is mainly related to the significant increase in humidity (WET) and decrease in dryness (NDBSI) in the area, highlighting the positive environmental effect of the naturally occurring water pits in the area. The secondary protection zone was protected before development, and the area contains 25 distinctive and accessible quarry ruins. Of the three classes of areas, the secondary protected area has the highest value-added vegetation (NDVI). Although the construction of the park caused a legacy of some bare rock and bare soil areas, and park construction increased the hard site area to some extent, the dryness index (NDBSI) within the area decreased compared to the original mining, indicating a decrease in the area of bare soil. The damage and impact of the subsequent construction on the environment was greatly lessened, including the ecological benign development around the relics. The tertiary protection zone was developed first and then protected, and the area contains eight quarry relics with poor landscape effects. The region directly used the hardened area of bare soil and bare rock in the original site for park construction. Its ecological quality was minimally enhanced, mainly caused by the negative effect of the significant increase in the heat level (LST) in the area, and its change was consistent with the increasing intensity of human activities in the area.

Overall analytical indicator relationships in the RSEI evaluation model had positive first principal component loadings for vegetation (NDVI) and humidity (WET) and the opposite for dryness (NDBSI) and heat (LST) [36]. Among the four indicators, dryness (NDBSI) had the largest load value and showed an increasing trend between 2014 and 2021, and its mean value showed a significant decrease between 2014 and 2021, indicating that in the study area, dryness had the greatest and continuously increasing impact on the RSEI [21]. The change in index had a significant impact on the improvement of the remote sensing ecological index in the region and played an important driving role [33]. On the one hand, it is clear that the original bare rock and bare soil areas had a negative impact on the location and surrounding ecological environment [37,38]; on the other hand, it also reflects the applicability of the RSEI in the evaluation of mine ecological quality.

The strategy of graded zoning, site-specific and moderate development and protection restored the ecological quality of the area and drove the economic development of the surrounding area on the basis of preserving the original appearance of the industrial ruins, which provides a practical reference for the ecological restoration path of the abandoned mine sites [39]. This study provides a practical reference for the ecological restoration of abandoned mine sites. However, the ecological restoration idea of turning the mine into a park is closely related to a number of water pits formed spontaneously in Tongluo Mountain Mine Park due to natural effects [40]. It can be said that water pits are the result of natural action. The construction of the local mine park was not only for the protection of the mining relics but also for restoration of the ecology in the area and for protection of this special landscape of human–nature coproduction [41]. The construction of the local mining park was not only for the protection of the mining relics but also for the restoration of the ecology in the area and the protection of this special landscape of humans and nature. Therefore, the construction of Tongluo Mountain Mine Park had its own special characteristics, and the ecological management of the mine should be planned according to the local conditions and should not be blindly imitated.

The existing evaluation index system still needs to be improved. Based only on the visualization results of the RSEI, the ecological quality grades of the area of bare soil caused by mining and the area where roads, flooring and buildings are located in the new park are both extremely poor, and it is difficult to distinguish the influence of both on the ecological quality of the area. The reason for this is that the dryness index was obtained by inversion of the mean values of the building index and bare soil index, which to a certain extent lessens the negative effect of the surface bare soil index on the ecological quality evaluation of the rare earth mining area and makes it difficult to visualize the regional impact of different features on the ecological environment. The related research combines the idea of scale in landscape ecology to propose a remote sensing ecological index based on moving windows, which optimizes the accuracy of the calculation of the RSEI in the case of more mixed regional feature types [42], which is applicable to similar special cases, such as bare earth land in mining areas and construction land transposition. The evaluation model can be optimized by introducing geographic probes [34], Google Earth Engine cloud computing platform [43] and other indicators. How to refine more targeted ecological evaluation indicators with this study needs further research. Meanwhile, the degradation of ecological quality in the park is closely related to human activities [21]. The degradation of ecological quality in the park is closely related to human activities. However, the construction and development of rural settlements around the park and the landscape construction in the park both play a positive role in the sustainable development of the regional economy and society and thus cannot be used as a basis for denying the ecological management effectiveness of the park. The model can be optimized by introducing human activity intensity [25] and nighttime lighting data [44]. The model can be optimized by introducing human activity intensity, nighttime lighting data and other indicators, or add a slope indicator to reflect the risk of erosion at the mine site.

## 5. Conclusions

The remote sensing ecological index is quantitatively constructed with four indicators, green, thermal, dry, and wet, and the objective and diverse characteristics can be applied well to the evaluation and evolutionary analysis of the ecological environmental status of regions with different degrees of development and treatment in the process of development-style management of mining areas. The ecological quality of the study area is improved comprehensively. The spatial and temporal dynamic change characteristics of the ecological environment at each quality level vary in the three-level relic reserve. The evaluation of ecological environmental quality in the three-level relic reserve of Tongluo Mountain Mining Park from 2014–2021 is analyzed as follows.

(1) Rare earth mines and other such over-exploited mines, the bare soil areas in the mines are mostly the plots where water and soil erosion occur, and emphasis should be placed on enhancing the vegetation cover of the bare soil areas to play a role in cooling, increasing humidity and enhancing the water and soil conservation capacity. In the face of the current situation of mining areas with different degrees of damage different focused soil and water conservation strategies should be adopted.

(2) The Tongluo Mountain mine pit restoration project improved the ecological quality in the region. As a product of resource-depleting collection behavior, mountain pits have increasingly significant negative environmental effects on mining areas. Applying the ecological remote sensing index (RSEI) to remote sensing monitoring of ecological quality and changes in pits enables rapid quantitative remote sensing evaluation of ecological changes in Tongluo Mountain Pit Park in Chongqing from 2014–2021, and the results of the study show that the overall ecological quality of the local area shows an upward trend. The three levels of protection indicate zoning treatment, and the treatment measures are adapted to local conditions, all of which improved the ecological quality within the region.

(3) The types of ecological changes in Tongluo Mountain pits are mainly divided into three categories: ecological conservation, resource diversion, and redevelopment. In terms of ecological enhancement, based on the principle of the three-level classification of pits, different utilization strategies are adopted for different levels of pits, and ecological conservation strategies are adopted inside and at the boundary of Pits 1–8 so that ecological quality is greatly improved in the remote sensing index. In terms of resource transfer, the mine pit with high landscape value is designed to become a mine park with experience, culture and education as the main focus; therefore, there is a small piece of land reclamation around the mine pit; thus, the ecological quality index is not improved as much as the area of the No. 1–8 mine pit, but it produces high landscape benefits and improves the overall conservation diversity of Tongluo Mountain. In terms of redevelopment, for the lower protection level of the mine area, we assumed the additional function of scenic tours and the secondary development of the surrounding green space to be used as a park-supporting facility arrangement place. In the original ecological base of the area, we did not overly pursue the rise of indicators but chose to provide more multifunctional support and, ultimately, to a certain degree, to improve the ecological quality based on the additional landscape facilities.

(4) For mine pit restoration according to the local conditions, the effect is good. Comparing the disposal measures for the three types of pits, the ecological enhancement as the main disposal measure contributes to the greatest ecological quality improvement benefit and ensures that the overall ecological quality of Tongluo Mountain Pit Park is continuously improved. In the comprehensive management system, resources are transferred and redeveloped, and different levels of development interventions are used in the secondary and tertiary areas with good ecological substrates so that both areas are upgraded to a good level, the diversity of pit protection measures is enhanced, and the scenic resources of the pits are efficiently used, completing the transformation from ecological resources to landscape resources.

## Figures and Tables

**Figure 1 ijerph-19-09750-f001:**
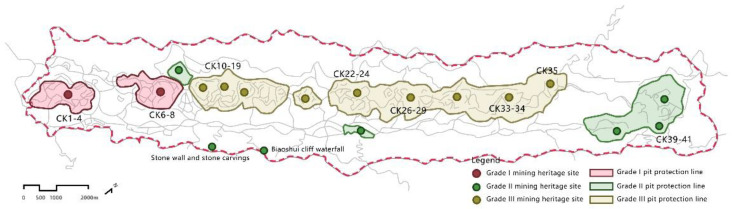
Protection and utilization planning of mining relics in Tongluo Mountain Mine Park.

**Figure 2 ijerph-19-09750-f002:**
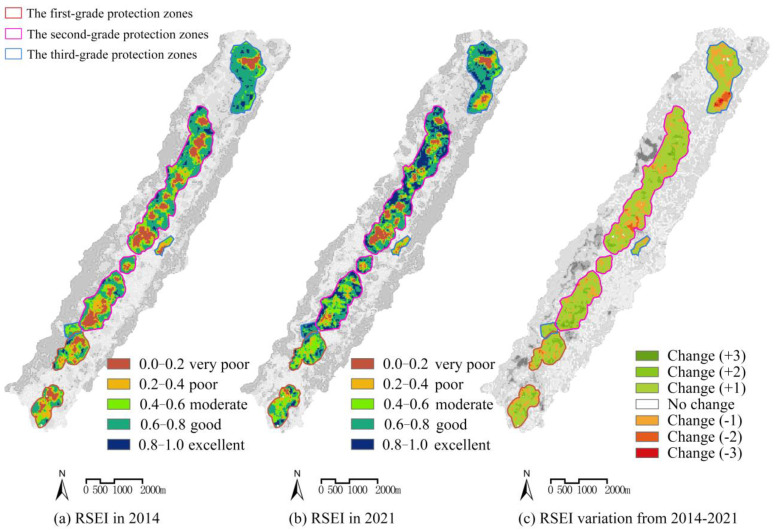
RSEI and its variation in Tongluo Mountain Mine Park from 2014 to 2021.

**Figure 3 ijerph-19-09750-f003:**
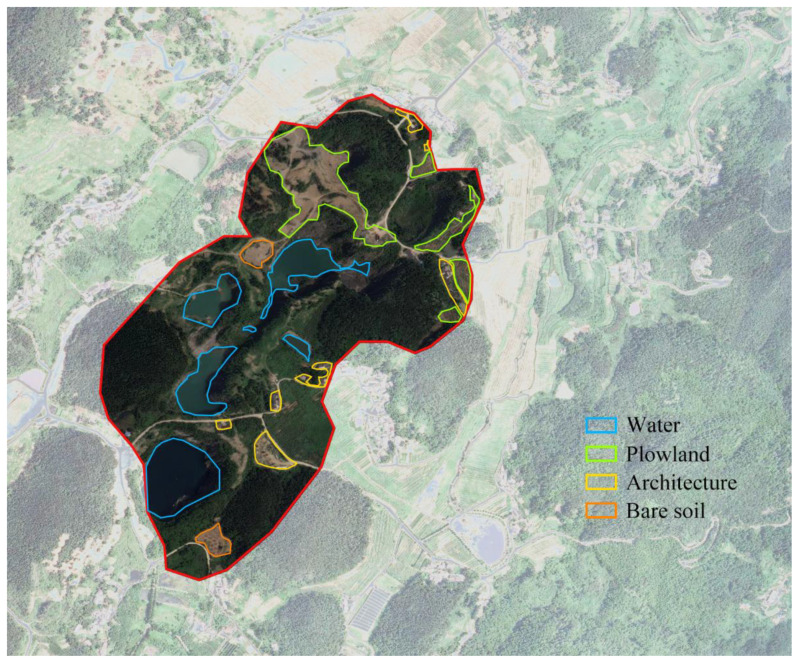
Aerial map with the sampling locations.

**Figure 4 ijerph-19-09750-f004:**
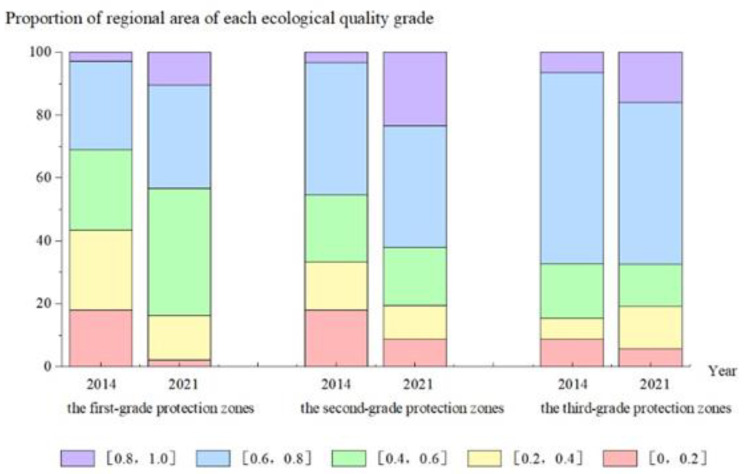
Percentage accumulation map of eco-environmental quality in different governance areas from 2014 to 2021.

**Figure 5 ijerph-19-09750-f005:**
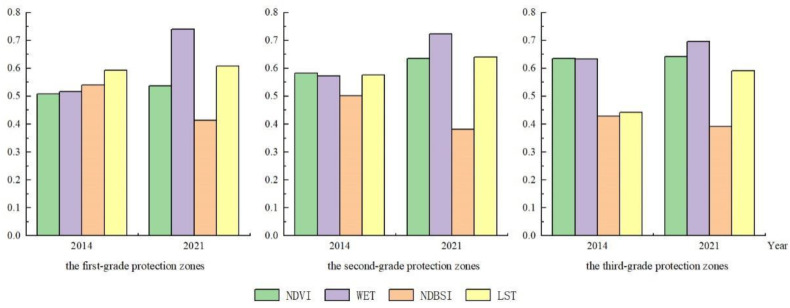
The mean changes in the normalized indices of ecological factors in the three zones.

**Table 1 ijerph-19-09750-t001:** Remote sensing ecological index calculation equation based on Landsat.

Year	Indicator	PC1	PC2	PC3	PC4
2014	NDVI	0.522858	0.525688	0.567416	−0.35821
	WET	0.521052	0.413196	−0.70196	0.25501
	NDBSI	−0.447353	−0.60023	−0.24506	−0.61607
	LST	−0.504975	−0.43892	0.353893	0.653537
	Eigenvalues	0.1008	0.0127	0.001	0.0006
	Eigenvalue Contribution Rate	87.59%	11.00%	0.87%	0.54%
2021	NDVI	0.521654	0.430223	0.692738	−0.2508
	WET	0.602344	0.343521	−0.71044	−0.12019
	NDBSI	−0.31315	−0.74006	0.008344	−0.59513
	LST	−0.516716	−0.38628	0.123759	0.753976
	Eigenvalues	0.1193	0.01	0.0021	0.0004
	Eigenvalue Contribution Rate	90.55%	7.60%	1.56%	0.29%

**Table 2 ijerph-19-09750-t002:** Area and proportion of each ecological level in the study area.

Quality Level	2014		2021		Growth during 2014–2021/km^2^
	Area/km^2^	Proportion/%	Area/km^2^	Proportion/%	
Very poor (0–0.2)	1.09	16.03%	0.48	6.98%	−0.62
Poor (0.2–0.4)	1.02	15.04%	0.81	11.87%	−0.22
Medium (0.4–0.6)	1.42	20.85%	1.42	20.89%	0.00
Good (0.6–0.8)	3.00	44.02%	2.73	40.06%	−0.27
Very good (0.8–1.0)	0.28	4.05%	1.37	20.19%	1.10
Mean	0.514385		0.604333		0.089948
Total area/km^2^	6.81				

**Table 3 ijerph-19-09750-t003:** Area and proportion of each ecological level in the study area.

Ecological Change	Rangeability	2014–2021	
		Area/km^2^	Proportion/%
Ecological Improvement	3.0	0.01	0.18%
	2.0	0.53	7.81%
	1.0	4.76	69.89%
Ecological Unchanged	0.0	0.00	0.00%
Ecological Degradation	−1.0	1.35	19.77%
	−2.0	0.14	2.12%
	−3.0	0.02	0.23%

**Table 4 ijerph-19-09750-t004:** Remote sensing ecological index of different governance areas and the overall area from 2014 to 2021.

	The Whole Area	The First-Grade Protection Zones	The Second-Grade Protection Zones	The Third-Grade Protection Zones
2014 RSEI	0.514385	0.455299	0.498493	0.604446
2021 RSEI	0.604333	0.576660	0.610487	0.610539
growth in value from 2014 to 2021	0.089948	0.121361	0.111994	0.006093
area/km^2^	6.81	1.19	4.11	1.51

**Table 5 ijerph-19-09750-t005:** Ecological factor index of different governance areas in 2014–2021.

Indicator	the First-Grade Protection Zones		the Second-Grade Protection Zones		the Third-Grade Protection Zones	
	2014	2021	2014	2021	2014	2021
NDVI	0.508773	0.537714	0.582730	0.635925	0.635969	0.641617
WET	0.517207	0.740030	0.572531	0.723445	0.632828	0.696893
NDBSI	0.540865	0.414093	0.500751	0.382151	0.428610	0.390973
LST	0.593211	0.608623	0.576721	0.640104	0.442797	0.592080

## Data Availability

The data that support the findings of this study are openly available at (https://earthexplorer.usgs.gov/, accessed on 25 February 2022).

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
