# Peer review of "Evaluation of Remote Sensing Ecological Index Based on Soil and Water Conservation on the Effectiveness of Management of Abandoned Mine Landscaping Transformation"

_ijerph, 2022, doi:10.3390/ijerph19159750_

Round 1

Reviewer 1 Report

The authors presented the results of applying the principal component method to calculate the " ecological index," whatever that means.
The article contains gross errors of both methodological, spelling, and semantic nature.
First, the citations in the IJERPH journal are formatted differently.
Second, the essence of the indices used and how they were obtained, as well as the connection with the "ecological" index, are not described. NDVI is a vegetation index, not a "greenness" index. How were the WET, LST, and NDSI indices calculated? The authors give indices such as PC1-4 in Table 1 - what is what? The principal component method is used to reduce the dimensionality of multivariate data and by itself cannot be a way to analyze the state of landscape disturbance - there are completely different metrics for that. Even assuming that the resulting RSEI somehow reflects the state of the underlying surface, the connection with reality is not tested here, since no verification work has been done. What is the correlation with the degree of real landscape disturbance? What factors contribute most to "restoration"?
There is no coordinate system in the drawings - it is impossible to estimate the real position of the research object.
Lines 424-426 are repetition. Google earth engine is a platform for calculating and analyzing RS data, it is not an indicator and it does not have any indicators, but it is possible to calculate them (lines 420-425).
I do not recommend the article for publication.

Author Response

Point(1):First, the citations in the IJERPH journal are formatted differently.

Responds: We have revised the article according to the citation format required in the journal. Thank you for your correction.

Point(2): Second, the essence of the indices used and how they were obtained, as well as the connection with the "ecological" index, are not described. NDVI is a vegetation index, not a "greenness" index. How were the WET, LST, and NDSI indices calculated?

Responds: We integrated the section on data pre pro cessing into Part 2.2 of the article and added references to illustrate the scientific basis for data pre processing. We also added the calculation methods and formulas for the four ecological indicators.

Point(3,4):The authors give indices such as PC1-4 in Table 1 - what is what?

Responds: PC1-4 in Table 1 represent the results of principal component analysis. In PC1, NDVI, representing vegetation, and Wet, representing humidity, are positive, indicating that they have a positive contribution to the ecosystem, while LST and NDBSI, representing heat and dryness, are negative, indicating that they have a negative impact on the ecosystem, which is consistent with the actual situation. In other characteristic components, these indicators fluctuate, which is difficult to explain. Therefore, compared with other components, PC1 has obvious advantages in that it can well integrate the information of each indicator and explain it reasonably.

Point(3,4):The principal component method is used to reduce the dimensionality of multivariate data and by itself cannot be a way to analyze the state of landscape disturbance - there are completely different metrics for that. Even assuming that the resulting RSEI somehow reflects the state of the underlying surface, the connection with reality is not tested here, since no verification work has been done. What is the correlation with the degree of real landscape disturbance? What factors contribute most to "restoration"?

Responds: Landscape restoration of mine is a ground restoration work with multiple measures, so it will have a certain impact on the local ground index. At the same time, landscape restoration is integrated by a number of work, and different restoration means and work have different impacts on the ground, so when different restoration means are used for different areas, they will have different ways and means of impact on indicators.

Point(5): Google earth engine is a platform for calculating and analyzing RS data, it is not an indicator and it does not have any indicators, but it is possible to calculate them (lines 420-425)..

Responds: The previous article may have been mistranlative. The idea was to introduce Google Cloud computing platform, geographic probes and other indicators to optimize evaluation model.

We deeply appreciate your consideration of our manuscript.If you have any queries, please don’t hesitate to contact me at the address below.

Best wishes.

Reviewer 2 Report

This manuscript uses remote sensing ecological index to assess the spatial and temporal differences in the ecological environment quality (RSEI) of tertiary relic reserves with different degrees of development and protection in the park. The construction of Tongluo Mountain Mine Park had its own special characteristics. Thus it may only be locally useful and lacks an international perspective. Although this work provides more information and attention for somewhere lacking research relatively, the innovative or research characteristics of this paper are insufficient.

There are some issues to modify:

1. Add 'Results showed that...' before Line 22.

2. A large number of repetitive statements appear in the manuscript, please carefully examine these errors. Location of repetition: Line40-43; Line50-52; Line53-57; Line63-66; Line61-62; Line140-143; Line176-180; Line424-426; Line430-434.

3. Line57: When you first mention it, it's best to pointed out what ’the four items of the RSEI ’ are.

4. Line98-113: Rearrange the logic and language to enhance the clarity of the manuscript. One classification is ‘Grade1-3’ based on the characteristics and functions of mining relics, and the other is classified into ‘primary protection zone’, ‘the second-grade protection zone’ and ‘a tertiary protection zone’ according to the radius. It can cause confusion.

5. ①Line157: 1RSEI0? ②Line178:NDSI?

6. Line200-202: Redundant. Or add the annotation under Figure 2.

7. With respect to the ground truthing, I would have preferred an additional map with the sampling locations instead of writing the results (Line220-227, for example).

8. Line239-240:I don't understand here. Why was the ecological quality of pits 1-8 decreased? Pits 1-8 are in the critical protected area, and they ought to be an increasing trend according to Table 4.

9. Introduction: â‘ Write in subsection. â‘¡The text suggests streamlining, many of which have been mentioned earlier. â‘¢There is somewhat little research literature on relevant content.

Author Response

Point(1-6):1. Add 'Results showed that...' before Line 22.

  1. A large number of repetitive statements appear in the manuscript, please carefully examine these errors. Location of repetition: Line40-43; Line50-52; Line53-57; Line63-66; Line61-62; Line140-143; Line176-180; Line424-426; Line430-434.
  2. Line57: When you first mention it, it's best to pointed out what ’the four items of the RSEI ’ are.
  3. Line98-113: Rearrange the logic and language to enhance the clarity of the manuscript. One classification is ‘Grade1-3’ based on the characteristics and functions of mining relics, and the other is classified into ‘primary protection zone’, ‘the second-grade protection zone’ and ‘a tertiary protection zone’ according to the radius. It can cause confusion.
  4. â‘ Line157: 1RSEI0? â‘¡Line178:NDSI?
  5. Line200-202: Redundant. Or add the annotation under Figure 2.

Responds: According to your suggestion, we have modified the repeated sentences and unclear parts of the article. The scientific problems faced and expected to be solved in this paper have been re-consolidated and the value of the research results has been increased.

Point(7): With respect to the ground truthing, I would have preferred an additional map with the sampling locations instead of writing the results (Line220-227, for example).

Responds: We added additional aerial maps to show the changes on the ground.

Point(8): Line239-240:I don't understand here. Why was the ecological quality of pits 1-8 decreased? Pits 1-8 are in the critical protected area, and they ought to be an increasing trend according to Table 4.

Responds: We have revised the description of this section. It is true that the protected area of pit 1-8 has an upward trend, but the periphery of the protected area has a downward trend due to the ecological construction trend

Point(9): Introduction: â‘ Write in subsection. â‘¡The text suggests streamlining, many of which have been mentioned earlier. â‘¢There is somewhat little research literature on relevant content.

Responds: We have revised the introduction based on your comments and added relevant literature.

We deeply appreciate your consideration of our manuscript.If you have any queries, please don’t hesitate to contact me at the address below.

Best wishes.

2022 07 20

Reviewer 3 Report

The article discusses a practical tool for the evaluation of the ecologial progress and discusses the result as made for a Mining Park in Chongqing.

In the introduction a lot of information is given dealing with the four ecological factors.

Heat index and dryness index are obtained from the inversion of surface temperature. Can you explain which way the inversion is done ? 

In line 60 is written 'The RSEI has more comprehensive and social criteria applicability', but this is not discussed further on in the article. Can it be explained which way the reader has to understand this ?

In my opinion this method compares momentaneous 'pictures' of the RSEI. Isn't it  important to evaluate also each of the factors in order to have a better understanding of  the evolution of the RSEI , instead of focusing only on the RSEI ?

Figure 1 is not clear.

In formule 1 the normalization of the index is discussed.  This facilitates the dimensionless processing, but the variation of the index will become higher. A relative small fluctuation within a narrow range can result in heigh normalised values. Because the four indicators are not normalised in the same way, it seems more difficult to distinguish the importance and the relative impact of each of the factors.

The discussion of the results is not clear. For example , an increment of 4.05 % to 20.29% is that something that could be expected, is that a large  increment ?  Is that spectacular ?  How can this increment be explained ? Is it possible to make a reference to an other type of evaluation or experience related to the evolution of an ecological progress ? 

This article discusses the results for one project based on the RSEI, but is it possible to give also a more global qualitative conclusion that expresses the improvement or the ecological progress. It should be also interesting to discuss the reasons for the change of the factors.  

Author Response

Point 1: The article discusses a practical tool for the evaluation of the ecologial progress and discusses the result as made for a Mining Park in Chongqing. In the introduction a lot of information is given dealing with the four ecological factors. Heat index and dryness index are obtained from the inversion of surface temperature. Can you explain which way the inversion is done ?

Response 1: We add Chapter 2.1.2. to introduce the calculation method of each index in detail. It describes inversion way we use.

Point 2: In line 60 is written 'The RSEI has more comprehensive and social criteria pplicability', but this is not discussed further on in the article. Can it be explained which way the reader has to understand this ?

Response 2: Most of the current remote sensing monitoring technologies are evaluated based on a single index, such as the use of NDVI in urban ecosystems. However, the evaluation of each indicator divided separately can only be one-sided explain the ecological characteristics of a particular aspect. RSEI integrates four ecological factors, compared with the single ecological factor, which is NDVI, it is more comprehensive to understand the change and improvement of ecological quality. Meanwhile Among the many natural factors reflecting ecological quality, greenness, humidity, heat and dryness are four important indicators closely related to human survival, and are also important factors for people to intuitively feel the merits of ecological conditions. Therefore The RSEI has a better performance in social criteria applicability.

Point 3: In my opinion this method compares momentaneous 'pictures' of the RSEI. Isn't it important to evaluate also each of the factors in order to have a better understanding of the evolution of the RSEI , instead of focusing only on the RSEI ?

Response 3: It is very important to evaluate also each of the factors, which can help us understand the change of ecological quality better. We carefully explain this in detail in Chapter 3.4.

Point 4: Figure 1 is not clear.

Response 4: We have replaced a clearer version.

Point 5: In formule 1 the normalization of the index is discussed. This facilitates the dimensionless processing, but the variation of the index will become higher. A relative small fluctuation within a narrow range can result in heigh normalised values. Because the four indicators are not normalised in the same way, it seems more difficult to distinguish the importance and the relative impact of each of the factors.

Response 5: We used PCA to process the data, which is a multi-dimensional data compression technique that selects a few important variables by linear transformation of multiple variables in order to distinguish the importance and the relative impact of each of the factors. However, the dimensions of the four indicators are not unified, and if PCA is directly calculated by them, the weight of each indicator will be unbalanced. Therefore, before principal component transformation, these indicators must be normalized first, and their dimensions should be unified between [0,1], which will not affect the importance and the relative impact of each of the factors.

Point 6: The discussion of the results is not clear. For example , an increment of 4.05 % to 20.29%

is that something that could be expected, is that a large increment ? Is that spectacular ? How can this increment be explained ? Is it possible to make a reference to an other type of evaluation or experience related to the evolution of an ecological progress ?

Response 1: We are using real historical data so this kind of massive increase cannot be expected quantitatively. In fact, it does look like a big increase in this program. Much of this massive growth is based on local work to regreen bare soil. This type of growth may also have the same trend in other ecological restoration processes with large bare soil area and severe damage.

Point 7: This article discusses the results for one project based on the RSEI, but is it possible to give also a more global qualitative conclusion that expresses the improvement or the ecological progress.     It should be also interesting to discuss the reasons for the change of the factors

Response 1: We totally agree with that. We also describe this change in the chapter 3.4 by analysing the four sub-indicators of RSEI, which explains the different tendencies and actual processes of ecological change.

Round 2

Reviewer 1 Report

The authors did not respond substantively to the comments, especially:
Responds: PC1-4 in Table 1 represent the results of principal component analysis. In PC1, NDVI, representing vegetation, and Wet, representing humidity, are positive, indicating that they have a positive contribution to the ecosystem, while LST and NDBSI, representing heat and dryness, are negative, indicating that they have a negative impact on the ecosystem, which is consistent with the actual situation. In other characteristic components, these indicators fluctuate, which is difficult to explain. Therefore, compared with other components, PC1 has obvious advantages in that it can well integrate the information of each indicator and explain it reasonably. - what is PC1, PC2, PC3, PC4 is still a mystery.
The fundamental question about "The principal component method is used to reduce the dimensionality of multivariate data and by itself cannot be a way to analyze the state of landscape disturbance - there are completely different metrics for that. Even assuming that the resulting RSEI somehow reflects the state of the underlying surface, the connection with reality is not tested here, since no verification work has been done." The authors do not verify their work in any way, they use methods that are not applicable to solve the tasks.

Author Response

Point(1):Responds: PC1-4 in Table 1 represent the results of principal component analysis. In PC1, NDVI, representing vegetation, and Wet, representing humidity, are positive, indicating that they have a positive contribution to the ecosystem, while LST and NDBSI, representing heat and dryness, are negative, indicating that they have a negative impact on the ecosystem, which is consistent with the actual situation. In other characteristic components, these indicators fluctuate, which is difficult to explain. Therefore, compared with other components, PC1 has obvious advantages in that it can well integrate the information of each indicator and explain it reasonably. - what is PC1, PC2, PC3, PC4 is still a mystery.

Responds: To calculate the RSEI, a principal components analysis (PCA) was adopted to identify the relative importance of each variable. During the study years, the contribute rate of PC1 were much higher than others, which indicated that PC1 has integrated most of the characteristics of all the variables, and thus PC1 was chosen to build the RSEI. Meanwhile, we have changed the expression of eigenvalue contribution rate in table in the manuscropt to avoid misunderstanding. While, PC2, PC3 and PC4 were not adpoted in the calculating of RESI due to their low contribution rates.

Table 1. Remote sensing ecological index calculation equation based on Landsat.

Year

Indicator

PC1

PC2

PC3

PC4

2014

NDVI

0. 522858

0. 525688

0. 567416

-0. 35821

WET

0. 521052

0. 413196

-0. 70196

0. 25501

NDBSI

-0. 447353

-0. 60023

-0. 24506

-0. 61607

LST

-0. 504975

-0. 43892

0. 353893

0. 653537

Eigenvalues

0. 1008

0. 0127

0. 001

0. 0006

Eigenvalue Contribution Rate

87. 59%

11.00%

0.87%

0.54%

2021

NDVI

0. 521654

0. 430223

0. 692738

-0. 2508

WET

0. 602344

0. 343521

-0. 71044

-0. 12019

NDBSI

-0. 31315

-0. 74006

0. 008344

-0. 59513

LST

-0. 516716

-0. 38628

0. 123759

0. 753976

Eigenvalues

0. 1193

0. 01

0. 0021

0. 0004

Eigenvalue Contribution Rate

90. 55%

7.60%

1.56%

0.29%

Point(2): The fundamental question about "The principal component method is used to reduce the dimensionality of multivariate data and by itself cannot be a way to analyze the state of landscape disturbance - there are completely different metrics for that. Even assuming that the resulting RSEI somehow reflects the state of the underlying surface, the connection with reality is not tested here, since no verification work has been done." The authors do not verify their work in any way, they use methods that are not applicable to solve the tasks.

Responds: In this study, we use RSEI to evaluate the ecological quality for an ecosystem. The principal component method was used to calculate RSEI from the four component indicators as mentioned above. The RSEI has been widly used to evaluate regional ecological quality [1-4]. Changes of biomass, humidity, dryness and land surface temperature caused by a disturbance of ecosystem will eventually be detected by RSEI. Hence, we identify the disturbance process based on the change of RSEI.

We deeply appreciate your consideration of our manuscript.If you have any queries, please don’t hesitate to contact me.

Best wishes.

References

  1. Xu, H.; Wang, M.; Shi, T.; Guan, H.; Fang, C.; Lin, Z. Prediction of ecological effects of potential population and impervious surface increases using a remote sensing based ecological index (RSEI). Ecol. Indic. 2018, 93, 730–740.
  2. Hu, X.; Xu, H. A new remote sensing index for assessing the spatial heterogeneity in urban ecological quality: A case from Fuzhou City, China. Ecol. Indic. 2018, 89, 11–21.
  3. Rikimaru, A.; Roy, P.S.; Miyatake, S. Tropical forest cover density mapping. Trop. Ecol. 2002, 43, 39–47.
  4. Xu, H. A new index for delineating built-up land features in satellite imagery. Int. J. Remote Sens. 2008, 29, 4269–4276.

Reviewer 2 Report

The revised manuscript is significantly improved. Authors have responded my concerns, thus there aren't further questions.

Author Response

We deeply appreciate your consideration of our manuscript. If you have any queries, please don’t hesitate to contact me.

Best wishes.